

# Perfluoroalkyl acids in sediment and water surrounding historical fire training areas at Barksdale Air Force Base

Rebecca S. Wilkinson[1], Heather A. Lanza[1], Adric D. Olson[1], Joseph F. Mudge[1], Christopher J. Salice[2] and Todd A. Anderson[1]

[1] Department of Environmental Toxicology, Texas Tech University, Lubbock, TX, United States of America
[2] Environmental Science and Studies, Towson University, Towson, MD, United States of America

Corresponding author
Todd A. Anderson,
todd.anderson@ttu.edu

## ABSTRACT

Perfluoroalkyl acids (PFAAs) are environmentally persistent components of surfactants that consist of fully fluorinated carbon chains and a terminal sulfonate or carboxylate polar head moiety. Due to their unique amphiphilic properties, PFAAs are used in the manufacturing of products such as aqueous film forming foams (AFFF). There is cause for concern for PFAA contamination resulting from runoff and groundwater infiltration of AFFF that were used during fire training. This study analyzed water and sediment samples that were collected over a 13-month sampling period from bayous upstream and downstream of two former fire training areas located near Barksdale Air Force Base (BAFB); the occurrence and magnitude of PFAAs supported an aquatic ecological risk assessment of potential impacts of PFAAs at the site. Liquid chromatography coupled with mass spectrometry was used for determination of 6 PFAAs listed under the third Unregulated Contaminant Monitoring Rule (UCMR 3). Total PFAA concentrations in surface water and sediment samples ranged from 0 (ND) $-7.1$ ng/mL and 0 (ND) $-31.4$ ng/g, respectively. Perfluorooctanesulfonic acid (PFOS) and perfluorooctanoic acid (PFOA) were the predominant PFAAs detected. In general, perfluorosulfonates were quantified more frequently and at higher concentrations than perfluorocarboxylates. The perfluoroalkyl chain length of PFAAs also showed significant influence on PFAA concentrations when analyzed by Spearman's rank correlation analysis. Some contamination we observed in surface water and sediment samples from reference locations could be a result of local runoff from the use of commercial products containing per- and poly-fluoroalkyl substances (PFAS), but AFFF appears to be the primary source given the close proximity of the historical fire training areas.

## INTRODUCTION

Per- and poly-fluoroalkly substances (PFAS) are a class of chemicals that have long half-lives and the ability to bioconcentrate, bioaccumulate, and biomagnify (*Conder et al., 2008*; *Anderson et al., 2016*). This class of chemicals, especially PFAS containing long perfluoroalkyl chains, are of increasing concern (*U.S. EPA, 2019*). Due to their unique

amphiphilic properties, PFAS are used in the manufacturing of products such as fire fighting foams, semiconductors, non-stick cookware, and waterproof clothing.

The first aqueous film forming foam (AFFF) was invented by Tuve, Spring, and Jablonski of the U.S. Naval Research Laboratory in the early 1960s and was patented in 1966 (*Alm & Stern, 1992*). One of the leading manufacturers of AFFFs was 3M; their perfluorinated surfactants were created through electrochemical fluorination. Electrochemical fluorination is a process that produces fully fluorinated compounds, such as PFOS (*Place & Field, 2012*). Other companies, such as National Foam and Ansul, used foams containing fluorotelomeres that were created through a process called telomerization (*Place & Field, 2012*). Although fluorotelomers are not fully fluorinated compounds, they have the potential to degrade to perfluorinated compounds, such as PFOA (*Wang et al., 2009*). On May 16th of 2000, 3M announced a voluntary phase out process of PFOS-based AFFFs due to 3M data that was reported to the Environmental Protection Agency (EPA) that found PFOS to accumulate and persist in humans and the environment (U.S. EPA 2000). Runoff and groundwater contamination from historical AFFF use raises concern for environmental contamination (*East, Anderson & Salice, 2021*).

Previous studies have evaluated concentrations of PFAS resulting from use of AFFFs. For example, *Moody et al. (2002)* began monitoring concentrations of PFAS in surface water and fish from Etobicoke Creek (Toronto, ON) the day after an accidental release of AFFFs at L.B. Pearson International Airport in June 2000. Surface water was collected over the course of 3 weeks after the spill and fish were collected in June 2000 and January 2001. Maximum concentrations (2,210 µg/L PFOS, 2,260 µg/L PFHxS, and 11.3 µg/L PFOA) were detected 1 day after the spill and had decreased to less than 5 µg/L 20 days after the spill (*Moody et al., 2002*). Monitoring of Etobicoke and Spring Creeks was continued for 9 years after the large AFFF spill in 2000 and results were published by *Awad et al. (2011)*. Concentrations of PFOS in water samples from the location closest to the spill decreased from 2003 to 2009 from 690 ng/L to 290 ng/L, respectively. However, sediment samples taken from this same location retained the same concentration of PFOS from 2003 to 2009 (13 ng/g) (*Awad et al., 2011*). Since our study was completed, there have been additional evaluations of surface water PFAS concentrations tied to historical AFFF use. Rather than list all of those studies here, we call the reader's attention to a recently published summary of the known PFAS surface water concentrations for the U.S. (*East, Anderson & Salice, 2021*).

The objective of this study was to determine the extent of PFAS contamination in water and sediment at Barksdale Air Force Base (BAFB) located in Bossier City, Louisiana in support of an aquatic ecological risk assessment (*Lanza et al., 2017*; *Salice et al., 2018*). BAFB has been an extremely active military installation and also is host to a wide variety of outdoor recreational opportunities. Historically, fire-training activities took place on the base, and there were concerns regarding the potential for contamination of surrounding areas with PFAS. Figure 1 is an aerial image of BAFB and provides an overview of where samples were obtained (locations identified in Table 1). A previous investigation had identified shallow PFAS contamination at both former Fire-Training Areas (FTAs) and a trace-level mass flux to Cooper Bayou; these flow patterns supported hypotheses concerning

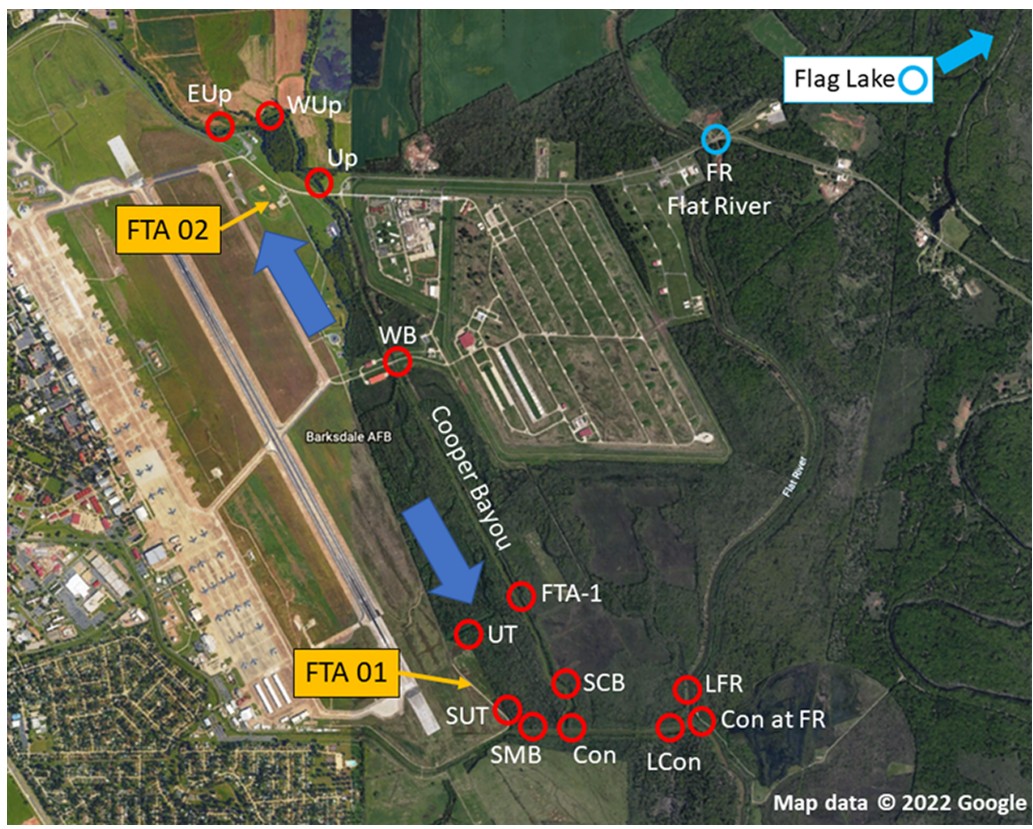

**Figure 1 Sampling locations at Barksdale Air Force Base.** Yellow arrows identify approximate locations of Former Fire Training Areas (FTA), large blue arrows represent general flow of groundwater below FTA, and red circles are where samples (sediment, water) were obtained. Blue circles represent on-installation reference locations that were outside the Cooper Bayou watershed, but may be influenced by other runoff. Map data © 2022 Google.

where PFAS would be detected in surface waters and sediments. However, because PFAS have been used in a wide variety of commercial products, environmental contamination could occur from a variety of sources. We, therefore, sampled two additional locations (Flat River and Flag Lake) on BAFB but distant from the FTAs. At the time of the study, there were 6 PFAS listed under the third Unregulated Contaminant Monitoring Rule (*U.S. EPA, 2012*); those analytes were the focus of our research project.

## MATERIALS AND METHODS

### Reagents

All standards, mass-labelled surrogates, and mass-labelled internal standards were purchased from Wellington Laboratories (Guelph Ontario Canada). Analytical standards of perfluoro-n-heptanoic acid (PFHpA), perfluoro-n-octanoic acid (PFOA), perfluoro-n-nonanoic acid (PFNA), potassium perfluoro-1-butanesulfonate (L-PFBS), sodium perfluoro-1-hexanesulfonate (L-PFHxS), and sodium perfluoro-1-octanesulfonate
**Table 1** Site names, abbreviations, and UTM coordinates (corresponding to Fig. 1).

| Site type | Location name | Abbreviation | UTM |
|---|---|---|---|
| Locations of Concern | Extreme Upstream | EUp | 15S 437476 3598161 |
| | West of Upstream | WUp | 15S 437723 3598348 |
| | Upstream | Up | 15S 0437799 3598273 |
| | Weapons Bridge | WB | 15S 0438513 3596688 |
| | Fire Training Area-1 | FTA-1 | 15S 0439395 3594985 |
| | Upper Tributary | UT | 15S 439186 3595100 |
| | South Upper Tributary | SUT | 15S 439505 3594490 |
| | Upper Tributary | UT at SMB | 15S 439539 3594417 |
| | South Cooper Bayou | SMB | 15S 0439555 3594443 |
| | South Mack's Bayou | SMB | 15S 0439497 3594410 |
| | Confluence | Con | 15S 0439587 3594415 |
| | Lower Flat River | LFR | 15S 440340 3594517 |
| | Confluence at Flat River | Con at FR | 15S 440367 3594448 |
| | Lower Confluence | LCon | 15S 440293 3594431 |
| Reference Locations | Flat River | FR | 15S 0440300 3598123 |
| | Flag Lake | FL | 15S 0444743 3596607 |

(L-PFOS) were obtained in methanol (MeOH). Surrogates were perfluoro-n-[1,2-$^{13}$C$_2$]hexanoic acid (MPFHxA), N-ethyl-d5-perfluoro-1-octanesulfonamide (d-N-EtFOSA-M), and perfluoro-n-[1,2-$^{13}$C$_2$]decanoic acid (MPFDA), also in methanol. Internal standards were (1) perfluoro-n-[1,2,3,4-$^{13}$C$_4$]octanoic acid (MPFOA) for PFHpA, PFOA, and PFNA, (2) sodium perfluoro-1-[1,2,3,4-$^{13}$C$_4$]octanesulfonate (MPFOS) for PFBS, PFHxS, and PFOS, and (3) N-methyl-d3-perfluoro-1-octanesulfonamide (d-N-MeFOSA-M). All organic solvents used in this project were LC/MS grade. Nitrogen and argon were UHP grade.

## Collection and extraction of water samples

Methods for water collection were adapted from EPA Method 537 (*Shoemaker, Grimmett & Boutin, 2009*). For this study, water samples, GPS coordinates, and water quality parameters were collected upon arrival at each location; our sampling included locations of concern (LOC) and reference locations (RL) (see Table 1). Samples consisted of approximately 500 mL of surface water collected in a 1-L polypropylene container which contained 0.5 g of water preservative (Trizma®, pH 7.4, Sigma-Aldrich). Each bottle was labeled with the sampling location and date and was transported back to the laboratory on ice.

Field reagent blanks (FRBs) were prepared in the laboratory by filling a 1-L polypropylene bottle with approximately 500 mL Milli-Q® water (MQH$_2$O), sealed, and transported to the field site. Upon arrival at the field site, the reagent water was poured into an empty 1-L polypropylene bottle, labeled as the FRB, and transported on ice back to the laboratory with the other field samples.

All samples were extracted within one week of collection date. Extraction at the laboratory was conducted by spiking an aliquot (250 mL) of the water samples or 250 mL of MQH$_2$O (lab blank) with 5 µL of 10 µg/mL surrogate mixture (SUR) and then passing the water

through solid phase extraction (SPE) cartridges (Bond Elut-LMS, 500 mg ×six mL, Agilent Technologies). Samples were eluted with five mL of MeOH, concentrated to 0.25 mL under nitrogen, and then brought to a final volume of 0.5 mL with an 80:20 MeOH:MQH$_2$O solution. Samples were then filtered using 2-mL centrifuge tube filters (0.22 $\mu$m, cellulose acetate, Corning Incorporated) at 7,000 rpm for 1 min. An aliquot of 196 $\mu$L was transferred to a 250-$\mu$L liquid chromatography (LC) vial along with 4 $\mu$L of 5 $\mu$g/mL internal standard mixture (IS).

## Collection and extraction of sediment samples

Sediment samples from Barksdale Air Force Base (BAFB) were collected from at least one foot away from the bayou bank and placed in a polypropylene jar. Each container was labeled with the sampling location and date. Samples were transported on ice to the laboratory.

All samples were extracted within one week of collection date. After arrival to the laboratory, approximately 60–70 g of each sediment sample was weighed and set in a vacuum hood to air dry. After at least 24 h of drying time, sediment was reweighed and a dry weight was recorded. The sediment was homogenized, placed in jars, and spiked with 5 $\mu$L of 10 $\mu$g/mL SUR. Enough MeOH (20–50 mL) was added to each sample to cover the sediment completely. Sample jars were placed on a shaker table at a speed of 120 rpm for 5 h. The MeOH extract was syringe filtered (0.2 $\mu$m, cellulose acetate, GE Whatman) into a 15-mL polypropylene centrifuge tube. Samples were concentrated to 0.25 mL under nitrogen and then brought back to a final volume of 0.5 mL with an 80:20 MeOH:MQH$_2$O solution. The samples were then filtered using 2-mL centrifuge tube filters (0.22 $\mu$m, cellulose acetate, Corning Incorporated) at 7,000 rpm for 1 min. An aliquot of 196 $\mu$L was transferred to a 250-$\mu$L LC vial along with 4 $\mu$L of 5 $\mu$g/mL IS.

## Instrumental analysis

The analytical methods for PFAS determination were adapted from EPA Method 537 (*Shoemaker, Grimmett & Boutin, 2009*). Analytes were separated using a Thermo Scientific liquid chromatograph equipped with a C-18 SecurityGuard$^{TM}$ column (2 × 2.1 mm) and a Gemini$^®$ NX-C18 analytical column (75 × 2 mm, 3 $\mu$m particle size). A Thermo Scientific Accela 1250 pump was operated at 300 $\mu$L/min using a mobile phase consisting of 20 mM ammonium acetate (A) and 100% MeOH (B). The elution gradient started at 60% A and 40% B and held for 1 min. The gradient was then ramped to 10:90 A:B over 24 min and held for 7 min. The gradient was then ramped to initial conditions (60:40% A:B) over 0.10 min and then held for 4.9 min. The sample injection volume was 30 $\mu$L.

A Thermo Scientific TSQ Quantum$^{TM}$ Access MAX triple quadrupole mass spectrometer was used to detect PFAS analytes. The electrospray ionization source (H-ESI, Thermo Scientific) was operated in negative ion mode with capillary needle voltage set to −3 kV and cone gas flow at 98 L/hr. The flow of nitrogen desolvation gas was set to 1,100 L/hr at a temperature of 350 °C.

## Quality assurance, quality control, and analyte quantification

Isotopic labeled surrogate compounds were fortified in each sample prior to the extraction process to estimate the percent recovery of analytes. In addition to estimating instrumental errors, isotopic labeled ISs were used to quantify PFAS analytes in environmental samples. A linear range was determined by a 5-point calibration curve consisting of concentrations ranging from 10 ng/mL to 250 ng/mL. The same standard vials used for the calibration curve were injected after every 4–5 environmental samples to serve as quality controls (continuing calibration check, CCC). Calculated PFAS concentrations were based on analyte:IS ratios and the linear equation produced by the standard calibration curve; they were not adjusted for extraction efficiency (surrogate recovery). Each water sample batch was evaluated for IS responses (within 70%–140% of the most recent CCC and within 50% of the average area measured during initial analyte calibration).

The lowest standard of 10 ng/mL was determined to be the limit of quantification (LOQ) based on repeated measures. A concentration of half the LOQ was substituted for samples with detectable concentrations below the LOQ. The limit of detection (LOD) was determined to be one ng/mL. Half of the LOD was substituted for the concentration of samples determined to be below the LOD when needed for statistical analysis purposes.

## Statistical analysis of data

R statistical software (version 2.11.1) was used to compute all statistics on data collected for this study. Shapiro–Wilk's test and Bartlett's test were used to determine if data sets were normally distributed and of equal variance, respectively. Non-normal data sets were subjected to transformations but often resulted in no improvement. Therefore, Spearman's rank correlation analysis was used to compare trends between 2 variables. Due to uneven data sets, a list-wise comparison was used for the Spearman analysis and paired deletions occurred if either the concentration or parameter were absent.

## RESULTS AND DISCUSSION

### Sediment grab samples

A total of 55 sediment samples were collected from locations of concern (LOC) at Barksdale Air Force Base during August and November of 2013 as well as March, early May, late May, June, July, and September of 2014. The majority of PFAS were above the detection limit, but not above the reporting limit. Perfluorosulfonates were detected more frequently than perfluorocarboxylates at levels above the limit of quantification. PFAS that were quantified in sediment grab samples were PFOS, PFHxS, and PFHpA. PFOS was quantified in 65% of sediment grab samples from the LOC and resulted in the highest average concentration ($2.75 \pm 5.28$ ng/g, $n = 36$). PFHxS was quantified in 27% of sediment grab samples from the LOC with an average concentration of $1.47 \pm 2.18$ ng/g ($n = 15$). The months of March and early May contained the highest average concentrations of PFAS quantified in sediment samples from the LOC ($5.55 \pm 8.13$ ng/g, $n = 13$ and 5.53 ng/g, $n = 1$, respectively). September was the only month where no PFAS were quantified in sediment samples from the LOC (Fig. 2). Late May was the only sampling period that resulted in quantifiable concentrations of PFASs at the reference locations (RL). However, these concentrations

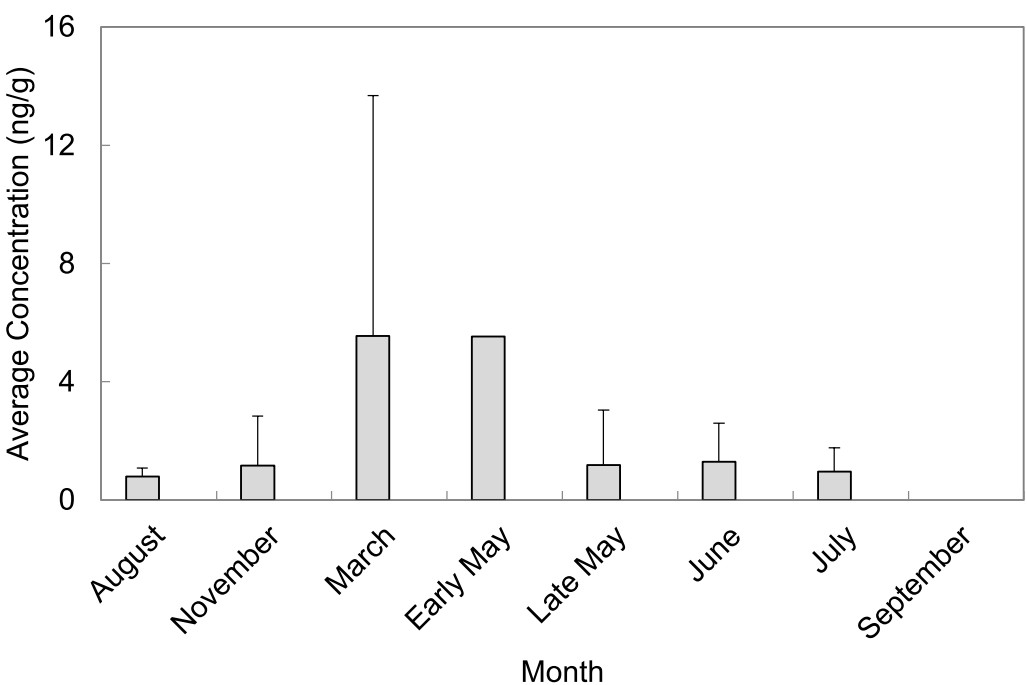

**Figure 2** Average concentrations (ng/g) of total PFAS per month in sediment grab samples from locations of concern (LOC) at Barksdale Air Force Base.

did not exceed concentrations quantified in samples collected from the LOC during the same sampling event.

Mean concentrations from all LOC and RL are presented in Fig. 3 (averaged across all sampling events). In general, observed sediment concentrations within Cooper Bayou were consistent with the hypothesis that the FTAs are point-sources of PFAS contamination. We did not expect sediment concentrations to fluctuate much seasonally. The intensity of some rain events common to the area undoubtedly moved sediments around and likely influenced apparent changes (decreases) in PFAS concentrations.

An increase in protonation of the sediment surface when the surrounding solution (water) is at a low pH leads to a strong electrostatic attraction between the sediment surface and anionic PFAS (*Zhao et al., 2014*). We observed a positive correlation with pH for PFBS, PFOS, PFHxS, and PFHpA, which is the opposite of the trend described by Zhao et al. However, the pH range (7.18–8.80) that was observed in the environment at BAFB may not have deviated enough to provide a true representation of the trend between PFAS and pH. Further, the relationship may also be influenced by the original PFAS composition of the AFFF(s) used in fire training.

Perfluoroalkyl chain length has a strong influence on sorption of PFAS to sediment (*Ahrens et al., 2011*). Although we observed positive correlations (rho = 0.24, $p < 0.005$, $n = 330$) between PFAS concentrations in sediment and molecular weight (which corresponds to perfluoroalkyl chain length), there are clearly some limitations to efficiently extracting PFAS from sediments (*Powley et al., 2005*; *Giesy et al., 2010*). It is likely that

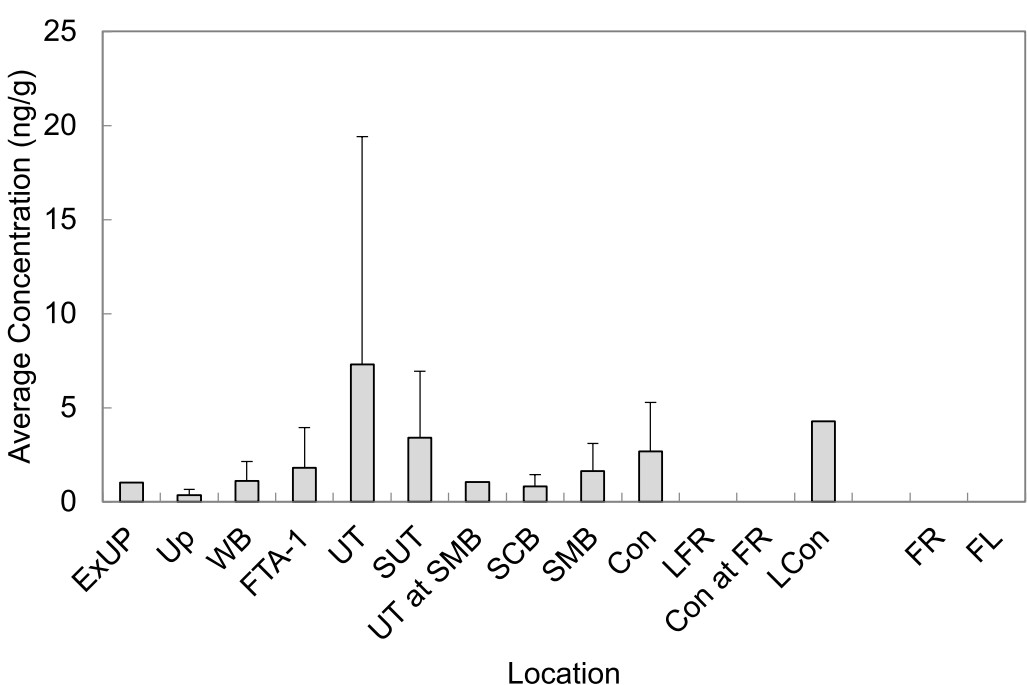

**Figure 3** **Average concentrations of all PFASs in sediment grab samples at all time points at Barksdale Air Force Base.** Reference locations (FR and FL) were nearly all < LOD.

concentrations of PFAS in sediment grab samples in the present study are underestimated given the low percent recovery of surrogates (average recovery for surrogates PFHxA-C13, PFDA-C13, and NEtFOSAA-d5 was 60 ± 10%, 38 ± 7%, and 19 ± 3%, respectively).

## Water grab samples

Surface water sampling during the months of August, November, March, early May, late May, June, July, and September resulted in the collection of 58 samples from LOC and 12 samples from reference locations. A total of 8 field reagent blanks (FRBs) and 8 lab blanks (LBs) were also analyzed for each month of water sampling. All water samples were fortified with surrogate analytes prior to extraction and resulted in average percent recoveries of >100 ± 32%, 95 ± 12%, and 98 ± 12% for PFHxA-C13, PFDA-C13, and NEtFOSAA-d5, respectively. The 58 water samples were analyzed in 7 batches following instrument calibration. Internal standard (IS) responses met QA criteria in six of seven batches for the sulfonate analytes, and seven of seven batches for the carboxylate analytes.

PFAS were quantified in every month of water collection. March, late May, and July had the highest average concentrations of total PFAS in water grab samples from the LOC (0.88 ± 1.07 ng/mL; 0.61 ± 0.87 ng/mL, and 0.54 ± 1.18 ng/mL, respectively). Average total PFAS concentrations were lower for the months of August, November, early May,

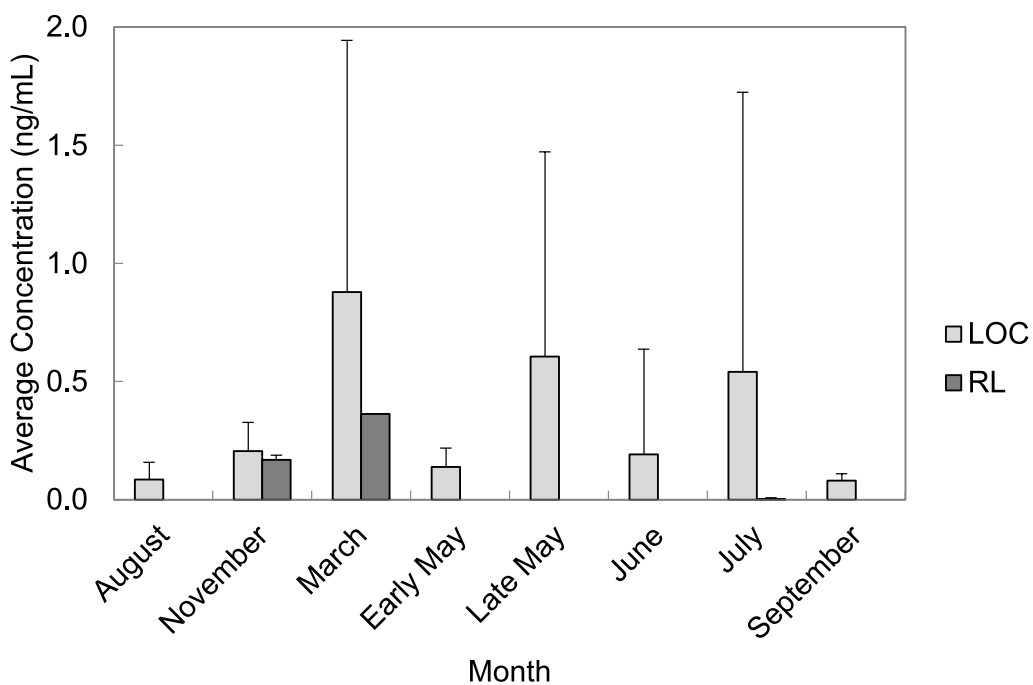

**Figure 4** Average concentrations (ng/mL) of all PFAS per month in water grab samples from locations of concern (LOC) compared to reference locations (RL) at Barksdale Air Force Base.

June, and September ranging from 0.01–2.76 ng/mL (Fig. 4). Rainfall data from the area suggested that the temporal trends we observed were related to increases in rainfall.

All LOC had at least one PFAS that was quantified. Locations Wup, UT, and SUT had the highest concentrations of all LOC sampled at BAFB (Fig. 5). Concentrations from these locations averaged 0.68 ng/mL ($n = 1$), 0.90 ± 1.99 ng/mL ($n = 42$), and 1.73 ± 1.77 ng/mL ($n = 8$), respectively. PFOS was observed with the highest average concentration (0.98 ± 1.38 ng/mL). However, PFHxS was the more frequently quantified (86%) and had the 2nd highest average concentration (0.61 ± 0.81 ng/mL). The observed water concentrations are consistent with those observed in sediments, again supporting the hypothesis that the FTAs are point-sources of PFAS contamination.

PFAS were detected at concentrations above the limit of quantification more frequently in water samples (63%) in comparison to sediment samples (16%), probably influenced to a large extent by the low extraction efficiencies for PFAS in sediment samples. The 3 sulfonates were the most frequently quantified and had higher average concentrations compared to any of the carboxylates. A study identifying PFAS components in 3M AFFF indicated only perfluorosulfonates were in surfactants produced between 1988 and 2001 (*Place & Field, 2012*). Therefore, perfluorosulfonates may be the predominant PFAS in water grab samples due to historical use of AFFFs produced by electrochemical fluorination; the quantification of perfluorocarboxylates could result from use of another product produced by a separate company. PFAS concentrations in water grab samples were also influenced by carbon

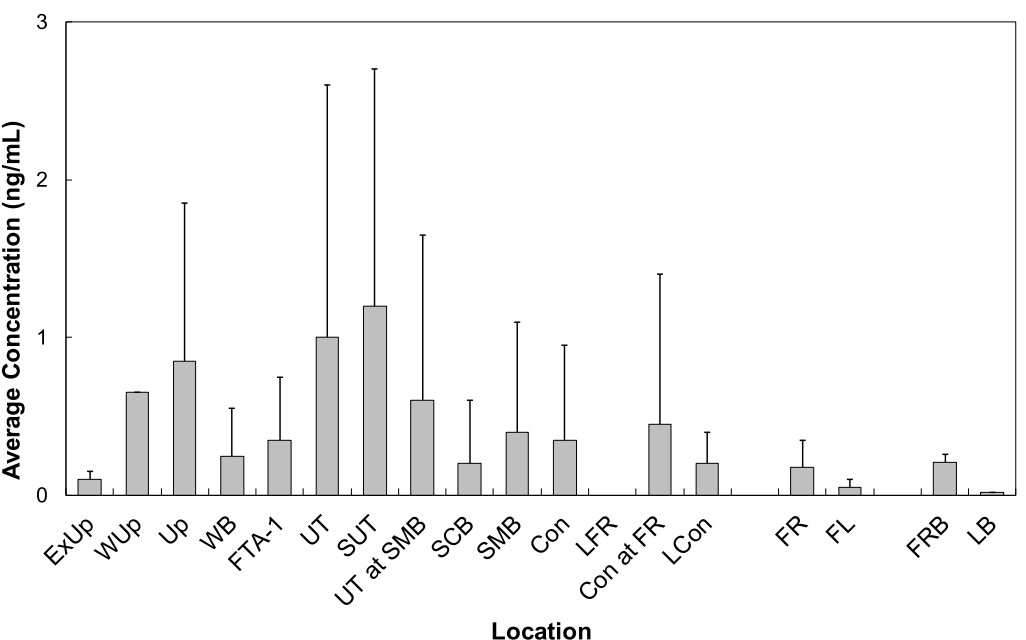

**Figure 5** **Average concentrations (ng/mL) of all PFASs at different locations within Cooper Bayou compared to reference locations and control samples.** PFAS concentrations were comprised mostly of PFOS.

chain length. We observed a significant positive correlation between concentration of perfluorosulfonates and corresponding chain length (rho = 0.36, $p < 0.005$, $n = 174$).

A previous study at BAFB revealed PFAS contamination at the 2 fire training areas (*Arcadis, 2013*); PFAS concentrations (PFOA: 30 μg/L; PFOS: 580 μg/L) collected from temporary groundwater wells just outside of the fire training burn pit were much higher than any of the concentrations detected in surface water samples in our study. However, when temporary shallow wells were sampled just north of the burn pit wells, concentrations decreased significantly (PFOA: 0.12 μg/L; PFOS: 1.1 μg/L) to levels more consistent with our surface water observations. Further, our PFAS observations in terms of quantities and mixtures is consistent with those reported in *East, Anderson & Salice (2021)*.

The average PFOS concentration (0.98 ± 1.38 ng/mL and 1.73 ± 1.77 ng/mL) detected in surface water samples at BAFB were similar to the PFOS concentration (0.69 ng/mL) detected in a water sample 3 years after an accidental spill in Toronto (*Awad et al., 2011*). However, the average PFOS concentration at BAFB was at least 3 times greater than the PFOS concentration (0.29 ng/mL) in a water sample collected 9 years after the accidental AFFF spill in Toronto (*Awad et al., 2011*).

## CONCLUSIONS

PFAS were detected in sediment and water grab samples. Overall, PFOS and PFHxS were the most frequently detected PFAS and were observed with the highest concentrations across all media types. There is most likely a continuous source of PFAS to Cooper Bayou

at BAFB from groundwater that was contaminated by historical use of AFFF. While some PFAS contamination could be a result of local runoff from the use of other PFAS-containing products, AFFF appears to be the primary source given the close proximity of the former fire training areas.

### Funding

This research was supported by the Air Force Center for Engineering and the Environment (AFCEE), Brooks Air Force Base. The funders had no role in study design, data collection and analysis, decision to publish, or preparation of the manuscript.

### Grant Disclosures

The following grant information was disclosed by the authors:
Air Force Center for Engineering and the Environment (AFCEE), Brooks Air Force Base.

### Competing Interests

Todd A. Anderson is an Academic Editor for PeerJ.

### Author Contributions

- Rebecca S. Wilkinson and Joseph F. Mudge performed the experiments, analyzed the data, prepared figures and/or tables, authored or reviewed drafts of the paper, and approved the final draft.
- Heather A. Lanza and Adric D. Olson performed the experiments, analyzed the data, authored or reviewed drafts of the paper, and approved the final draft.
- Christopher J. Salice and Todd A. Anderson conceived and designed the experiments, performed the experiments, analyzed the data, prepared figures and/or tables, authored or reviewed drafts of the paper, and approved the final draft.

### Field Study Permissions

The following information was supplied relating to field study approvals (i.e., approving body and any reference numbers):

Access to the field sampling locations was provided by Barksdale Air Force Base.

### Data Availability

The raw measurements are available in the Supplementary Files.

### Supplemental Information

Supplemental information for this article can be found online at http://dx.doi.org/10.7717/peerj.13054#supplemental-information.

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
