# Peer review of "Perfluoroalkyl acids in sediment and water surrounding historical fire training areas at Barksdale Air Force Base"

_PeerJ, doi:10.7717/peerj.13054_

## Round 0.1 · original submission · Major Revisions

In order to be considered for publication, the manuscript must be improved to a large extent.

Reviewer 1 ·

Basic reporting

Scientific writing needs to be improved
Literature should be updated
Data should be reanalysed, the presentation of averages of all chemicals removes all of the important information

Experimental design

Why was a temporal assessment chosen?
If this study was in support of an ecological assessment it would have been intersting to compare these results to what was found in the ecological assessment, as not all PFASs accumulate. An assessment of the contamination gradient with increasing distance from the point source could have been done.

Validity of the findings

There is nothing novel about the study in its current presentation

Additional comments

The study reports PFAA concentrations in sediment and surface water around an army base. It seems like a very extensive sampling campaign was conducted that could have been used for a really worthwhile assessment of contaminant distribution, however, the results presented are very limited, have major drawbacks in the analytical and add no new information.
Abstract: it promises more than what is behind the study. If the ecological assessment mentioned in the abstract will play no part in the entire manuscript it may just be better to leave it out. Where are the results of the spearman-rank correlation mentioned in the abstract? How do the authors come to the conclusion that local runoff from the use of commercial products (other than AFFF) contribute to the contamination observed?
Introduction: the introduction reads like a thesis, is very descriptive and cites publications from 10+ years ago. Some information contained in the introduction is not correct; i.e. PFOS is not the only PFAS listed in the Stockholm Convention, PFOA has also been listed and others are under review for inclusion. Up to date information on the use of AFFF in the US could have been included, as I am sure that has also changed in the last 10 years.
Materials and methods: the analytical standards (mass-labelled surrogates) selected are somewhat ill fitted, although it is impossible to tell which surrogate was used for quantification of which native. Usually matching standards are used, i.e. mass-labelled PFHxS for the analysis of native PFHxS and so on. Having at least some matching mass-labelled standards for such a limited standard suite of PFAAs is an absolute must. If no matching mass-labelled standards are available one would at least use those with the closest mass and the same headgroup as a surrogate.
Results and discussion: The results are presented as averages across the months and lump together all PFAAs analysed. Why was this done? All of the interesting information is lost that way and one is left with a bar chart of sediment averages that show no differences between the months with such high standard deviations. Sediments are also quite heterogeneous a variation is already expected between sampling events. The limitations of the sediment extraction method are more likely a results of the ill-fitted standards rather than the extraction method itself. The authors could have used the data it much better ways to describe the flow of chemicals away from the point source. For example changing ratios of chemicals, both in sediment and in water, which would also be related to the chemical properties.
Conclusions: a conclusions section usually reports the main findings of the study on hand. Here, the results are compared to a study from 2011.

·

Basic reporting

This manuscript presents some interesting data on current PFAS concentrations at a historically AFFF-contaminated site. The manuscript is well-written and fairly clear, and the amount of background/context given was sufficient. However, some of the driving hypotheses are unclear. The manuscript could be improved by focusing in some more on what can be learned from this dataset - it seems the data point towards an AFFF point source coming from the FTAs, and that a comparison to older data from the site indicates that concentrations have not decreased. These are interesting findings, and the manuscript could be improved by focusing in on these points.

It is unclear what the driving hypothesis is for looking at average concentrations every month at all locations. We expect and observe significant variation site-to-site, so it might be better to focus on specific sites in Figures 1 and 3 to understand temporal variability.

Additionally, the results & discussion section could be improved by adding more depth - for example in lines 283-288, it is mentioned that a previous study found higher concentrations than those found in this study. This observation is mentioned briefly but no explanation is given as to what it means or why it might be significant. Delving into a comparison of previous measurements versus those here would be really interesting.

Other specific comments:

Line 83: missing word between "the" and "and"
Lines 100-101: here "PFASs" is used whereas before the authors used "PFAS" as the plural form - please check through the manuscript and make sure you are consistent with how you're handling PFAS acronyms.
Lines 224-225: here it is stated that concentrations at RLs were quantifiable in Late May, but in Figure 2 it is stated that all PFAS were <LOD at RLs.
Line 282: A correlation is observed between chain-length and water concentration. I think the discussion of this needs to be expanded upon for it to really be a useful observation. Also, the authors should make it clear that while there are many data points from different samples, there are few compounds (so few chain-lengths) being used for the regression. What's the meaning of this regression? Could it be due to the original AFFF composition?

Experimental design

Methods:

For the most part, accepted methods were used for sample preparation and analysis. However, I have some concerns about methodology, particularly related to surrogate recoveries, and whether variation in sample-to-sample recovery could be affecting the observations made.

General comments:

As I'm sure the authors are aware, the low SURR concentrations for sediment extraction are of concern, as they are outside of what would usually be considered acceptable. Because the IS's used for quantitation do not correspond to the target analytes, that makes it more difficult to understand how robust the data is...

One way to address this issue to some extent and increase confidence in the data would be to describe the SURR recovery variability. While for the most part, the authors appear to have been quite diligent about QC, I saw no mention of collection of replicates, or the variability in replicate samples. This would be really helpful in understanding if seasonal changes observed are actually significant. Additionally, if low SURR recoveries are consistent, it increases confidence in observed trends.

Specific comments:

Lines 110-120: Typically when quantifying PFAS, you want to be very careful to quantify using the corresponding labeled compound whenever possible to avoid matrix effects and differences in ionizability that are difficult to predict. I found it very unusual that sulfonamide standards were used here to quantify PFAAs, and I think this should be addressed in the manuscript. Furthermore, the authors should specify which IS was used to quantify which compound. Choice of IS can affect quantitation.

Validity of the findings

The manuscript is transparent in providing raw data. Additional confidence could be gained if a presentation of replicates was given, and if IS quantitation was explained further and uncertainties due to non-ideal standards being used were explicitly described.

The conclusions would be strengthened if more detail was added to the discussion, particularly in relation to temporal comparisons (both seasonal and comparison to previous study).

Additional comments

Thank you for your work on this interesting study. You have produced some data that will be of interest to others studying AFFF-impacted sites. My comments on how to further improve the manuscript are detailed in the previous sections.

---

## Round 0.2 · accepted · Accept

The revised version of the manuscript was improved in a satisfactory level.

·

Basic reporting

The manuscript has been improved based on my initial round of review and I have no further suggested edits.

Experimental design

no comment

Validity of the findings

no comment

Additional comments

no comment